# Knockout of DDM1 in *Physcomitrium patens* disrupts DNA methylation with a minute effect on transposon regulation and development

**Ofir Griess**[ORCID][☯], **Katherine Domb**[☯], **Aviva Katz, Keith D. Harris, Karina G. Heskiau, Nir Ohad**\*, **Assaf Zemach**[ORCID]\*

School of Plant Sciences and Food Security, Tel-Aviv University, Tel- Aviv, Israel

☯ These authors contributed equally to this work.
\* assafze@tauex.tau.ac.il (AZ); niro@tauex.tau.ac.il (NO)

**Data Availability Statement:** BS-seq and RNA-seq data are available from the GEO database under the accession number GSE198693.

## Abstract

The Snf2 chromatin remodeler, DECREASE IN DNA METHYLATION 1 (DDM1) facilitates DNA methylation. In flowering plants, DDM1 mediates methylation in heterochromatin, which is targeted primarily by MET1 and CMT methylases and is necessary for silencing transposons and for proper development. DNA methylation mechanisms evolved through-out plant evolution, whereas the role of DDM1 in early terrestrial plants remains elusive. Here, we studied the function of DDM1 in the moss, *Physcomitrium (Physcomitrella) patens*, which has robust DNA methylation that suppresses transposons and is mediated by a MET1, a CMT, and a DNMT3 methylases. To elucidate the role of DDM1 in *P. patens*, we have generated a knockout mutant and found DNA methylation to be strongly disrupted at any of its sequence contexts. Symmetric CG and CHG sequences were affected stronger than asymmetric CHH sites. Furthermore, despite their separate targeting mechanisms, CG (MET) and CHG (CMT) methylation were similarly depleted by about 75%. CHH (DNMT3) methylation was overall reduced by about 25%, with an evident hyper-methylation activity within lowly-methylated euchromatic transposon sequences. Despite the strong hypomethylation effect, only a minute number of transposons were transcriptionally activated in *Ppddm1*. Finally, *Ppddm1* was found to develop normally throughout the plant life cycle. These results demonstrate that DNA methylation is strongly dependent on DDM1 in a non-flowering plant; that DDM1 is required for plant-DNMT3 (CHH) methylases, though to a lower extent than for MET1 and CMT enzymes; and that distinct and separate methylation pathways (e.g. MET1-CG and CMT-CHG), can be equally regulated by the chromatin and that DDM1 plays a role in it. Finally, our data suggest that the biological significance of DDM1 in terms of transposon regulation and plant development, is species dependent.

**Funding:** This work was supported by the Israeli Centers for Research Excellence Program of the Planning and Budgeting Committee, Israel Science Foundation (757/12), Israel Science Foundation (1636/15), and the European Research Council (679551) to A.Z, and Israel Science Foundation (767/09) to N.O. The funders had no role in study design, data collection and analysis, decision to publish, or preparation of the manuscript.

**Competing interests:** The authors have declared that no competing interests exist.

## Introduction

DNA methylation is catalyzed by DNA methyltransferases (DNMTs) with unique affinities and targeting mechanisms. In plants, DNA methylation is divided **into** three sequence contexts, CG, CHG, and CHH (H = A, C, or T) [1]. In flowering plants, CG methylation is maintained by MET1, CHG methylation by CMT3, and CHH methylation by CMT2 or DRM2 [2–6]. To access and methylate the DNA, DNMTs are assisted by Snf2 chromatin remodelers [7–10]. The most robust Snf2 remodeler for methylation in plants is DDM1 [11]. Mutations in DDM1 in flowering plants, such as in *Arabidopsis*, reduced methylation in all three sequence contexts suggesting its general importance for the methylation of distinct DNMTs [2,12]. DDM1 is especially important for DNA methylation of nucleosomal DNA of heterochromatic transposable elements (TEs), there it was suggested to counteract the chromatin compaction determined by the linker histone H1 or histone variant H2A.W [2,13–15]. To date, the roles of DDM1 in DNA methylation, TE regulation, and plant development were investigated in four flowering plants, namely *Arabidopsis*, tomato, rice, and maize [2,16–19]. In association with their methylation effect, *ddm1* mutations in flowering plants were found to be crucial for the silencing of TEs as well as for proper development; thousands of TEs were activated in *ddm1* mutants, which have shown pleiotropic to lethal developmental phenotypes [2,16,17,20,21].

While DNA methylation is found in all land plants, its genomic profile and enzymology can be different among species [22,23]. For example, in comparison to *Arabidopsis* and other flowering plants, in the moss *P. patens* non-CG methylation is dependent on a CMT and DNMT3 rather than on CMT2, CMT3, and DRM2 [24,25]. DNMT3 is an ortholog of animal DNMT3 that has been lost during the evolution of flowering plants, and its CHH methylation activity at heterochromatic sequences was replaced by the flowering specific CMT, CMT2 [25]. DRMs are plant-specific homologs of DNMT3 and function together with the RNA directed DNA methylation (RdDM) pathway in methylating euchromatic TE sequences [2,3]. DRMs exist in *P. patens*, however, have only a trivial role in DNA methylation [25]. Essentially, in *Arabidopsis*, CMT2 and DRM2 are responsible for 70% and 30% of CHH methylation, respectively [2,3], whereas 95% of CHH methylation is mediated by DNMT3 in *P. patens* [25]. Another unique difference between *P. patens* and many flowering plants is the robust CHG methylation in the former, which is found at a similar level as its CG methylation. For example, CG and CHG methylation in *Arabidopsis* TEs is approximately 80% and 50%, respectively, whereas, in *P. patens* it is about 80% for both contexts.

DNA methylation in *P. patens* regulates TE expression and development [26]. Symmetric and asymmetric methylation function redundantly in silencing TEs [26]. Additionally, CHG methylation in *P. patens* has a stronger silencing effect than CG methylation, and a null mutant for CHG and CHH methylation develops abnormally [26]. While CG and CHG methylation can be separately eliminated in *P. patens* (including that of CHH methylation), efforts to generate a null *P. patens* mutant for both CG and CHG methylation failed to be accomplished, possibly due to lethality [26].

Accordingly, by studying the function of DDM1 in *P. patens*, we would elucidate its role in an early terrestrial plant, i.e., a non-flowering one. More specifically, we will discover the effect of DDM1 on DNA methylation mediated by a distinct DNMT (i.e., DNMT3), its contribution to a methylome mediated primarily by heterochromatic methylases (i.e., with a trivial RdDM activity), its relative influence on CG versus CHG methylation (that are equally methylated in *P. patens*), as well as the impact of symmetric methylation on TE control and *P. patens* development (assuming CG and CHG will be significantly reduced).

To study the role of DDM1 in *P. patens*, we first knocked out its single DDM1 gene by homologous recombination, which proved to be viable. We then profiled *Ppddm1* methylome,

transcriptome, and morphology. Our data found PpDDM1 to be crucial for the three DNA methylation sequence contexts. Specifically, both symmetrical methylation contexts (CG and CHG) were affected similarly and significantly more than asymmetric CHH methylation (reduction of 75% vs 25%, respectively). Despite the methylation effect, TE expression and development of *Ppddm1* were hardly altered. Overall, our data substantiate the role of DDM1 in DNA methylation in plants evolution, show the dependency of a plant DNMT3 on DDM1, demonstrate the relative effect of DDM1 on CG and CHG methylation, as well as find that a massive disruption of CG and CHG methylation can be insufficient to activate TEs nor to alter plant development.

## Results

### *Physcomitrium patens* DDM1 knockout develops normally

To investigate the biological role of DDM1 in early terrestrial plants, we aimed to mutate it in *Physcomitrium patens*. A DDM1 gene has been previously identified in *P. patens*, named PpDDM1 (Pp1s65_183v6.1; [27]). To verify that PpDDM1 is the only DDM1 gene in *P. patens* (some plant species encode for multiple genes [16,17]), we generated a phylogenetic tree containing all *P. patens* Snf2 ATPases together with previously identified functional DDM1 orthologs from plants and mammals. This analysis located only PpDDM1 in the monophyletic DDM1 clade (Fig 1A). We confirmed this finding by generating a second phylogenetic tree containing *Arabidopsis* Snf2 remodelers (CHRs), together with PpDDM1 and its two closest homologs in *P. patens* (Pp3C11_18g and Pp3c73340). In this tree, PpDDM1 was placed inside the DDM1 clade, whereas Pp3C11_18g and Pp3c73340 were located together with the *Arabidopsis* CHR11 and CHR17 clade (Fig 1B), the latter belonging to the SWR1 complex that deposit H2A.Z in genes [28]. Altogether, these findings suggest that PpDDM1 is the only DDM1 gene in *P. patens*.

To knockout DDM1 in *P. patens*, we performed gene replacement using homologous recombination through the transformation of *P. patens* with the Zeocin resistance gene flanked by PpDDM1 sequences (Fig 1C). Following antibiotic selection, we identified a PpDDM1 knockout plant. Despite the identification of 609 differentially expressed genes (S1 File), morphological investigation of *Ppddm1* found it to be developed normally throughout the plant life cycle as well as in subsequent generations (Fig 1D–1H).

### DNA methylation is severely disrupted in *Ppddm1*

To check the role of DDM1 on DNA methylation in *P. patens*, we profiled the methylome of *Ppddm1* using whole genome bisulfite sequencing (WGBS). DNA methylation was substantially reduced in *Ppddm1* (Fig 2A). In transposons, which is the main target for methylation in *P. patens*, CG, CHG, and CHH methylation levels in *Ppddm1* were reduced by 76.7%, 77.8%, and 24.1%, respectively (Fig 2B). This result implies that the symmetric CG and CHG sites are both hypomethylated to a similar extent, which is substantially stronger than that of asymmetric CHH methylation. Meta analyses of methylation across TEs and chromosomes substantiated these findings (Fig 2C–2G). The similar hypomethylation effect in CG and CHG in *Ppddm1* is an interesting result, as these methylation sites are independently targeted and maintained in *P. patens* by a distinct type of methylases; CG sites are methylated by PpMET, whereas CHG sites by PpCMT [25]. CG and CHG methylation levels can be influenced by site density. Thus, to check for the relative hypomethylation effect of CG versus CHG, we next tested it within 50 bp windows with a similar number of CG and CHG sites. Within those sequences, CG and CHG methylation levels were similar in wild type and *Ppddm1* mutant (Fig 2H), with a median at zero for comparison between mCG and mCHG (CG minus CHG) in

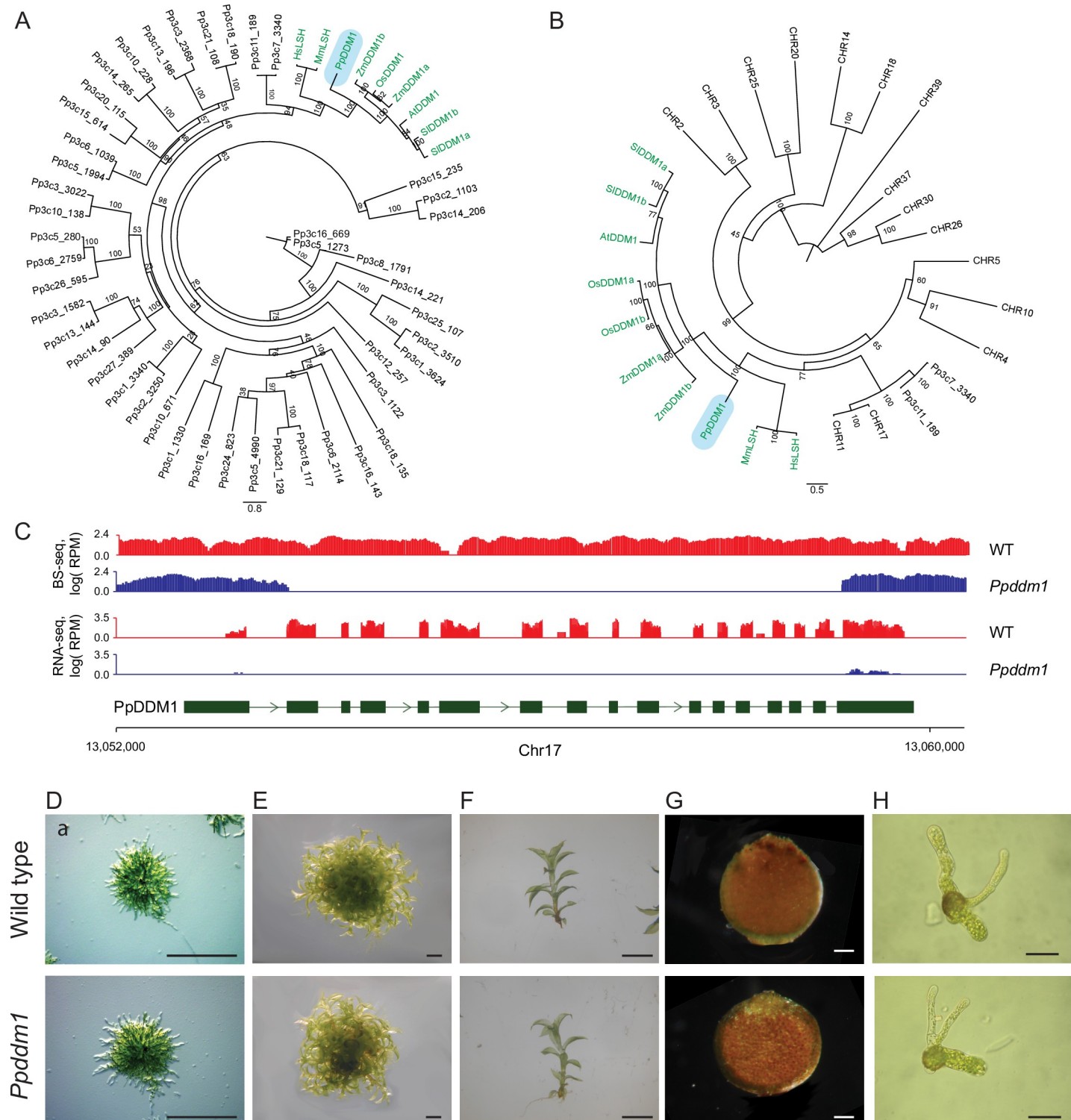

**Fig 1. *Physcomitrium patens* DDM1 knockout develops normally.** A. Phylogenic analysis of all *P. patens* SWI/SNF2 protein family members (including PpDDM1), and previously characterized DDM1 orthologues from flowering plants and mammals (LSH). The DDM1 clade is marked in green and PpDDM1 is highlighted in blue. B. Phylogenic analysis of A. thaliana SWI/SNF2 protein family members, PpDDM1, and additional DDM1 orthologs from flowering plants and mammals. DDM1 clade is marked in green and PpDDM1 is highlighted in blue. C. Coverage data from BS-seq and RNA-seq show the depletion of DDM1 sequences within the recombination region (Chr17:13053812-Chr:13059096). (D-H). Morphological phenotypic analysis of the wild type and *Ppddm1* plants. Protoplast-recovered protonematal tissue (D), gametophore tissue (E), single gametophore (F), sporophyte (G), and germinated spores of T1 (H). Scales for D/E, F/H, and G, are 2 mm, 100 uM, and 50 uM, respectively.

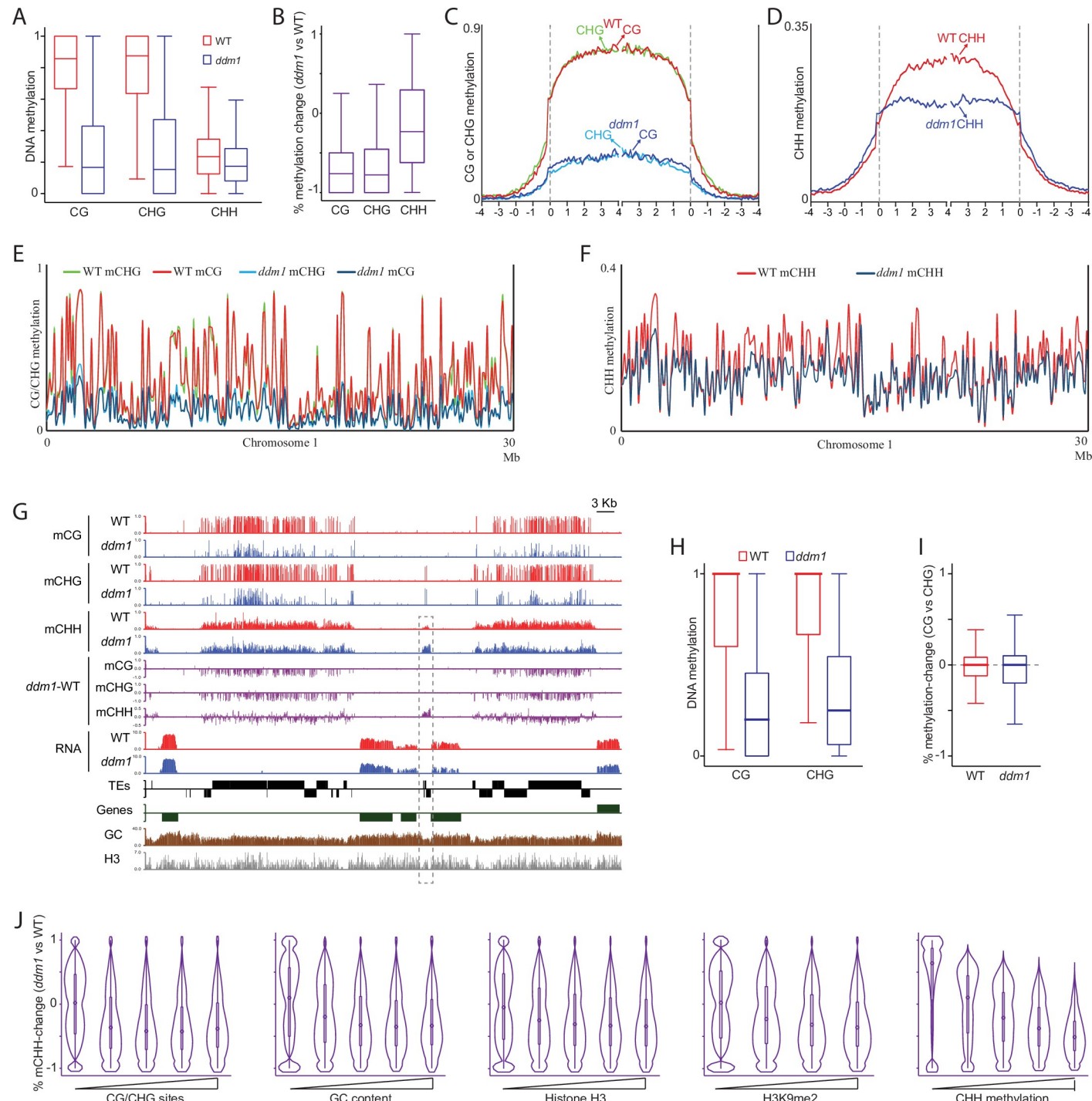

**Fig 2. DNA methylation in *Ppddm1*.** A. Box plots of DNA methylation (separated to CG, CHG, and CHH contexts) in wild type and *Ppddm1* in TEs. B. Box plots of percent methylation change in TEs between *Ppddm1* versus wild type calculated for 50-bp windows. C-D. Patterns of CG and CHG methylation (C) and CHH methylation (D) in wild type and *Ppddm1* across TEs. TEs were aligned at the 5′ or 3′ ends, and methylation was averaged within 50-bp intervals. The dashed lines represent points of alignment. E-F. Average of CG and CHG methylation (E) and of CHH methylation (F) in wild type and *Ppddm1* along *P. patens* chromosome 1 averaged in a 100-kb sliding window. G. A genomic representative snapshot of hypo-methylated TEs and a hyper-methylated TE (marked in dotted line) in *Ppddm1*. DNA methylation, methylation difference (*Ppddm1 minus* wild type), RNA level, GC content, and histone H3 are plotted in 50-bp windows. Genes and TEs oriented 5′ to 3′ and 3′ to 5′ are shown above and below the line, respectively. H. Box plots of CG and CHG methylation in wild type and *Ppddm1* within methylated 50-bp windows (>0.5 methylation in either of the samples) and with an equal number of CG and CHG sites. I. Percent-methylation change between mCG and mCHG in wild type and *Ppddm1* (i.e. WT-mCG minus WT-mCHG (red) and *ddm1*-mCG minus *ddm1*-mCHG (blue)), within windows with an equal number of CG and CHG sites. J. Violin plots of percent-methylation-change in 50-bp windows of CHH methylation in TEs between *Ppddm1* and wild type, separated based on the level (quantiles) of indicated attributes. Histone H3 and H3K9me2 data were derived from Widiez *et al.*, [29].

wild type and in *Ppddm1* (Fig 2I). Together, these results suggest that CG and CHG methylation are similarly affected by the deletion of PpDDM1.

In comparison to symmetric sites, which were consistently hypomethylated in *Ppddm1* (Fig 2B, 2C, 2E and 2G), asymmetric CHH sites showed some hyper-methylation effect in addition to its hypo-methylation (Fig 2B, 2D, 2F and 2G). By dissecting the methylation sequences to different chromatin features, we located the CHH hyper-methylation effect to euchromatic-TE sequences, i.e., regions depleted of GC content, nucleosomes, or the heterochromatic mark H3K9me2 (Fig 2J). CHH hyper-methylation was particularly negatively associated with CHH methylation in wild type, i.e. CHH sites with normal average methylation lower than 15% were mainly hyper-methylated, whereas CHH sites with normal average methylation between 23–47% were mostly hypo-methylated in *Ppddm1* (Fig 2J, rightmost panel). These results align with previous findings on the redistribution of CHH methylation in *ddm1* mutants [16,18].

## PpDDM1 knockout upregulated a small number of TEs

*Ddm1* mutants in flowering plants caused a substantial upregulation of hypomethylated TEs, i.e., more than 1000 upregulated TEs in *Arabidopsis*, tomato, and rice ([2,16,30]; Fig 3A). To check the effect of PpDDM1 on TE regulation, we profiled the transcriptome of *Ppddm1* via RNA-seq. A differential expression analysis identified only 35 TEs that were upregulated in *Ppddm1* (Fig 3A and S1 File). TE downregulation was negligible in either of the *ddm1* mutants (S1A Fig). Next, we checked for a meta-transcriptional effect of *Ppddm1*, i.e., on entire TE families rather than on single TEs. To do so, we quantified the total number of reads mapped to entire TE families (including multi-mapped reads). This analysis did not discover a considerable TE upregulation in *Ppddm1* (S1B Fig), thus substantiating the trivial transcriptional effect of *Ppddm1* also among non-unique TE sequences, such as of recently duplicated elements. *Ppddm1*-upregulated TEs were not more hypomethylated than unaffected TEs (Fig 3B), suggesting that not the level of hypomethylation triggered the expression in this group of TEs. Alternatively, we found that *Ppddm1*-upregulated TEs were depleted of methylation within their long terminal repeat (LTR) sequences in wild type plants (Fig 3C and 3D). Furthermore, *Ppddm1*-upregulated TEs were slightly expressed within their LTRs already in wild type (Fig 3D). Accordingly, these results suggest the minute effect of TE regulation in *Ppddm1* is focused on a small group of TEs that are predisposed to be expressed.

Such a result could have suggested that DNA methylation is not necessary to silence TEs in *P. patens*. However, mutations in the DNA methylases (DNMTs) in *P. patens* caused to upregulation of up to about 3600 TEs (Fig 3F) [26], that included most of upregulated TEs in *ddm1* (Fig 3F). Accordingly, the strong TE-upregulation in *Ppdnmt* mutants implies the importance of DNA methylation in the silencing of *P. patens* TEs. DNA hypomethylation in *Ppddm1* is different than any of that of the methylases mutants. For example, *Ppcmt* and *Ppmet* mutants were entirely, and almost exclusively, depleted in CHG and CG methylation, respectively [25]. In comparison, in *Ppddm1* both CG and CHG sites were strongly but also partially hypo-methylated (Fig 3B). At the CHH context, *Ppcmt* and *Ppddm1* had a similar complex effect, that is hypo and hyper-methylation in hetero- and eu-chromatic TE sequences, respectively [25]. When averaging the combined methylation level of all three methylation contexts (referred to as total methylation), *Ppddm1* had a lower total methylation level than *Ppcmt* (Fig 3E), implying that close to 100% reduction of methylation in CHG has a bigger effect on TE expression than a 75% reduction in both CG and CHH methylation. This result suggests a lack of a complete redundancy between CG and CHG methylation in the silencing of *P. patens* TEs and that the pattern of hypomethylation among the three methylation contexts, rather than the total methylation level, has a stronger influence on TE regulation.

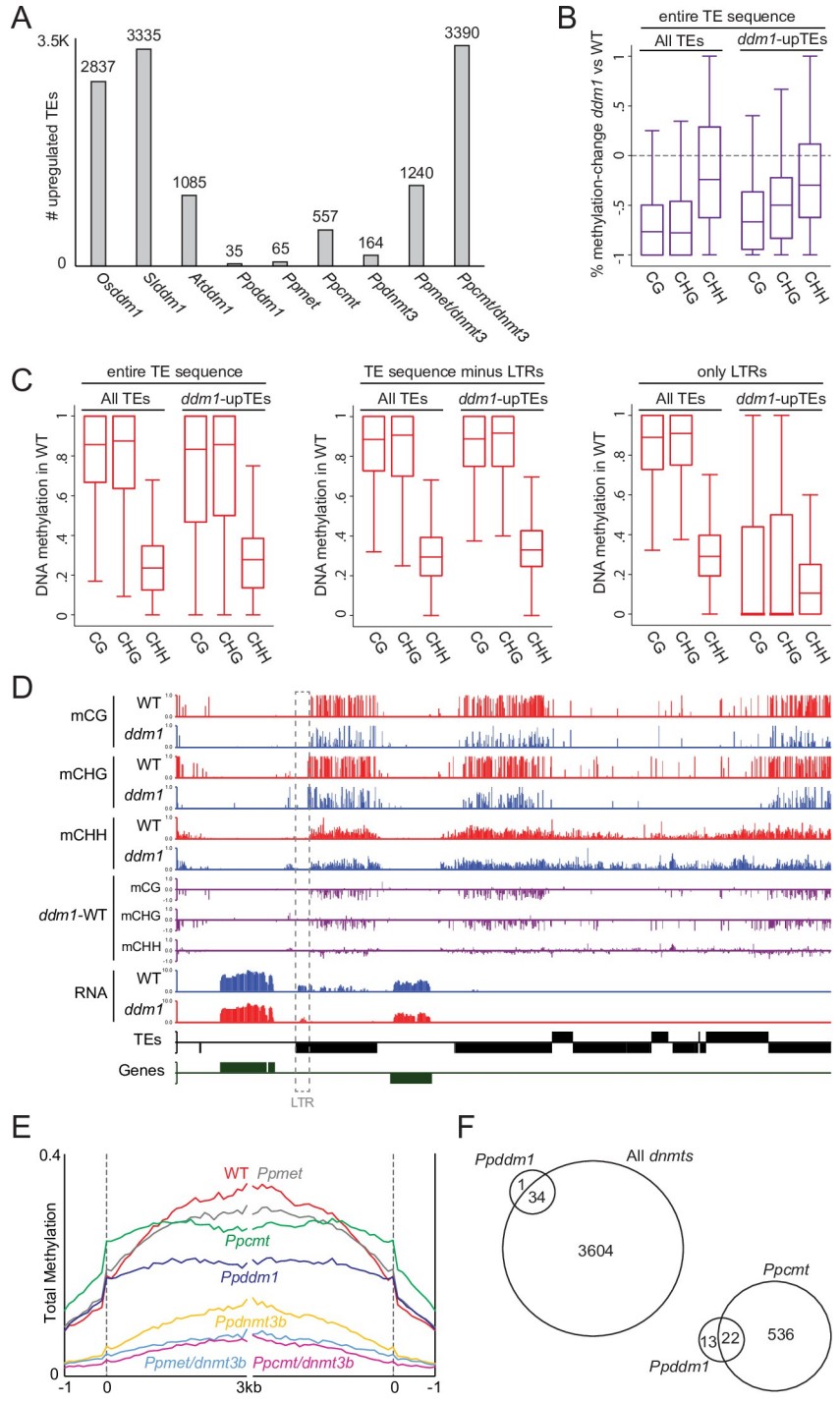

**Fig 3. PpDDM1 knockout upregulated a small number of TEs. A.** Number of upregulated TEs in indicated DDM1 and DNMT mutants. *Osddm1* and *Slddm1* are *Oryza sativa* and *Solanum lycopersicum* plants, respectively, that are each mutated in two DDM1 genes (*ddm1a* and *ddm1b*) [16,21]. At*ddm1* is an *Arabidopsis ddm1* mutant [2]. *P. patens met*, *cmt*, and *dnmt3* mutations are notated with the Prefix Pp [26]. *Osddm1*, *Slddm1*, and *Atddm1* data are from leaves or roots, and all *P. patens* samples were from protonema. **B.** Percent methylation change between *Ppddm1* and wild type in all versus *Ppddm1*-upregulated TEs. **C.** Wild type DNA methylation in all TEs versus *Ppddm1*-upregulated TEs across the entire TE sequence (left panel), internal TE sequences (middle panel), or only within long terminal repeat (LTR) sequences (right panel). **D.** A genomic representative snapshot of hypo-methylated upregulated and unaffected TEs in *Ppddm1*. The dotted line marks an LTR region that is already hypomethylated and expressed in wild type. **E.** Total methylation (CG, CHG, and CHH methylation combined) across TEs in wild type, *Ppddm1*, and *Ppcmt*. **F.** Venn diagram of upregulated TEs in *Ppddm1* and all *Ppdnmts* (left diagram) or *Ppcmt* (right diagram).

## Discussion

By investigating DDM1 in the moss *P. patens*, our data substantiate the role of DDM1 in mediating DNA methylation in plants. Similar to flowering plants, PpDDM1 is required for maintaining all three DNA methylation contexts in TEs. CHH methylation in flowering plants is mediated by DRMs and CMTs, where the activity of the latter is most dependent on DDM1. In comparison to flowering plants, CHH methylation in *P. patens* is mediated primarily by DNMT3, i.e., hardly by CMT or DRMs. Thus, the disruption of CG, CHG, and CHH methylation in *Ppddm1*, suggests that DDM1 is required by all types of plant methylases that function within heterochromatic TE sequences, that is MET1, CMTs, and DNMT3 enzymes. Nevertheless, the level of DDM1 dependency on particular methylases could be different. In the flowering plant *A. thaliana*, heterochromatic CHH methylation mediated by CMT2 is strongly dependent on DDM1, and to about a similar extent to MET1's CG methylation and CMT3's CHG methylation [2]. Here, we found CHH methylation in *Ppddm1* to be reduced three times less than CG and CHG methylation, suggesting that PpDDM1 is more crucial for symmetric versus asymmetric methylation, as well as that PpDNMT3 heterochromatic CHH methylation activity is less dependent on DDM1 than that of the *A. thaliana* CMT2's.

In flowering plants CG and CHG sites are usually methylated to a different level in transposons and are interdependent, CG methylation influences CHG methylation and vice versa [31–35]. Therefore, the relative role of DDM1 on CG methylation versus CHG methylation could have not been identified. In comparison to flowering plants, CG and CHG sites in *P. patens* TEs are methylated to a similar extent and are regulated independently, i.e. complete removal of CG methylation, as shown for *Ppmet*, hardly affects CHG methylation and vice versa, when *Ppcmt* is compromised [25,36]. Accordingly, profiling the methylome of *ddm1* in *P. patens* can reveal the relative role of DDM1 on CG and CHG methylation. Our data show that both CG and CHG methylation are being depleted by about 75% in *Ppddm1*, suggesting their similar dependency on DDM1. CMT and MET1 are distinct in their protein structure and mode of action. CMTs contain a chromodomain within their catalytic domain, which recruits them to heterochromatin by binding to H3K9me2 [37], whereas MET1s are targeted to chromatin via UHRF1/VIM proteins [38]. Accordingly, the similar extent of CG and CHG hypomethylation in *Ppddm1* illustrates the convergence of two distinct and independent methylation enzymatic pathways to be equally regulated by the chromatin remodeler DDM1.

Mutations of DDM1 in flowering plants caused massive upregulation of TEs associated with abnormal developmental phenotypes [2,16,18,30]. In comparison to published data in flowering plants, our data show that mutation of *ddm1* in moss, does not cause a substantial activation of TEs (at least not in the protonema tissue) nor an obvious developmental phenotype. This trivial transcriptional and developmental effect of *Ppddm1* suggests that the residual methylation is sufficient to silence TEs and maintain proper development. Furthermore, we show that total methylation of *Ppddm1* is lower than *Ppcmt*, while the latter has a greater effect on TEs. In comparison to *Ppddm1* which lost 75% of CG and CHG methylation, *Ppcmt* lost close to 100% of CHG methylation and 0% of CG methylation. These results support our previous finding that CHG methylation is a stronger TE-silencer than CG methylation [26], and imply that not the total amount of methylation is important to silence TEs but rather its distribution among the different methylation contexts. Accordingly, while the molecular effect of DDM1 on DNA methylation could be similar among different plants, its biological role can be different and is determined by the methylation landscape of the particular organism.

## Conclusion

Our results extend our knowledge of the role of DDM1 in plants. We show that DDM1 is necessary for any heterochromatic-DNA methylases, including for CHH methylation by plant DNMT3. We further show that separate methylation pathways can be similarly regulated by chromatin and DDM1. Finally, the trivial effect of *Ppddm1* on TE expression and plant development suggests that the biological effect of DDM1 is species dependent.

## Materials and methods

### Phylogenic reconstruction and motif finding

Protein sequences of DDM1/LSH were obtained using the Phytozome and Swissprot databases. *Arabidopsis thaliana* SWI/SNF2 members were found using the SMART and Eggnog databases. *Physcomitrium patens* SWI/SNF2 members were found using Pfam. Full protein sequences (S1 Table) were aligned by MAFFT using the BLOSUM62 algorithm [39] and trees were built using PhyML [40] (NNJ) with bootstrap value 100, and were illustrated using Figtree v1.4 [41]. Motifs were obtained by using MEME-suite under default options [42].

### Biological materials

Plant material: *Physcomitrium patens* line "Grandsen2004" was grown on either BCD, BCDAT, or germination media [43] in 16/8 hours of light to the dark regime at a light intensity of 10–50 μmol/m$^2$/s. For sporulation, the light regime was 8/16 hours light to dark at a light intensity of 10–50 μmol/m$^2$/s.

### Generation of *Ppddm1* knockout mutant and growth conditions

*Ppddm1* was obtained through the replacement of the gene coding region with Zeocin resistance antibiotic cassette through homologous recombination. Homologous flanking 5' (1068bp upstream to ATG site) and 3' (252 bp upstream to stop site and 536bp downstream to stop site) were amplified using KOD polymerase (Sigma-Aldrich) and cloned into a PMBL5-Zeo vector. Integration of flanking regions was validated using PCR. Linearized plasmid was introduced to moss via PEG-mediated transformation protocol [43]. After three days, the regenerated protoplasts were transferred to the BCDAT medium. After seven days the moss was subjected to selection using Zeocin (25ng\μl). After three weeks, surviving plants were subjected to PCR to validate the correct integration of the construct into the genome and gene replacement. This was done through primers targeting the endogenous gene sequence and primers targeting regions flanking the 5' and 3' homologous regions and Zeocin resistance cassette.

### BS-seq library preparation

Genomic DNA of 7-day old protonemata was extracted using the Plant II midi kit (Machery-Nagel) according to the manufacturer's instructions. About 1 μg of purified genomic DNA was fragmented by sonication, end-repaired, and ligated to custom synthesized methylated adaptors. Adaptor ligated libraries were subjected to sodium bisulfite treatment using the Methylation-Gold Bisulfite Kit (ZYMO) as outlined in the manufacturer's instructions. The converted libraries were then amplified by PCR, gel-purified to select the 350- to 400-bp size range, quantified using real-time quantitative PCR with TaqMan probe, and validated with a Bioanalyzer (Agilent). The libraries were amplified on cBot to generate the cluster on the flow cell (TruSeq PE Cluster Kit V3–cBot–HS, Illumina) and sequenced as a paired end on the HiSEq. 2000 System (TruSeq SBS KIT-HS V3, Illumina).

## BS-seq data analysis

The quality of BS-seq reads (S2 Table) was evaluated using FastQC v0.11.8 [44]. Trimming of low-quality reads was conducted using TrimGalore v0.6.0 [45]. Duplicated reads were removed using Prinseq-lite v0-20-4 [46]. Methylation data processing of qualified reads was performed as described previously [26]. Briefly, we converted all of the Cs in the "forward" reads (and in the scaffold) to Ts, and all of the Gs in the "reverse" reads and scaffold to As. The converted reads were aligned to the converted scaffold (*P. patens* v3.3) using Bowtie2 v2.3.4.1 [47], allowing two mismatches and multimapping reads with up to 10 hits. We then recovered the original sequence information and, for each C (on either strand), counted the number of times it was sequenced as a C or a T. Fractional methylation was calculated within a 50-bp sliding window for each sequence context separately (CG, CHG, or CHH) or all sequence contexts together (total methylation). The percent of methylation change was calculated by dividing the difference in methylation level between two samples by the level of methylation in the sample with the higher methylation level, e.g. (mCG in WT–mCG in *ddm1*)/mCG in WT if mCG in WT>mCG in *ddm1* and (mCG in *ddm1* –mCG in WT)/mCG *ddm1* if mCG in *ddm1*>mCG in WT. Meta-analysis of DNA methylation relative to TE edges was performed by aligning TEs at either their 5′ or 3′ ends and averaging methylation in a sliding 100 bp windows upstream and downstream of the point of alignment. Elements were included in the meta-analysis either when they reached the end of their sequence, another annotated element, or the indicated selected length. Box and density plots were generated on STATA v14. Public BS-seq raw data of *A. thaliana* (SRR771524, SRR771518) [2], *S. lycopersicum* (ERS2066779, ERS2066938) [16], and *O. sativa* (SRR3503137 SRR3503134) [21], were downloaded from the National Center for Biotechnology Information (NCBI) Sequence Read Archive (SRA) and processed as described above.

## RNA-seq library preparation

Three repetitions of total RNA were extracted from 7-day old protonema using the SV Total RNA Isolation System (Promega) and DNase treatment (Thermo Fisher Scientific) according to the manufacturers' instructions. The polyA fraction (mRNA) was purified from 500 ng of total input RNA, followed by fragmentation and the generation of double-stranded cDNA. Next, Agencourt Ampure XP beads cleanup (Beckman Coulter), end repair, A base addition, adaptor ligation, and PCR amplification steps were performed. Libraries were quantified by Qubit (Thermo Fisher Scientific) and TapeStation (Agilent). The libraries were sequenced as single-end reads on HiSEq 2500.

## RNA-seq data analysis

The quality of RNA-seq reads (S3 Table) was evaluated using FastQC v0.11.8 [44]. Trimming of low-quality reads, PolyA-tails, and adapter sequences was conducted using TrimGalore v0.6.0 [45]. The postprocessed reads were aligned to the genome using STAR v2.7.0.f [48]. Multimapped reads (<100) have been randomly assigned. RNA-seq stats are summarize in For differential gene and TE expression, we used the Limma R package [49]. Genes and TEs with a read count below 10 were omitted. Genes and TEs with 2-fold change and p-val<0.05 were considered as differentially expressed and used in downstream analyses. Public RNA-seq raw data of *A. thaliana* root tissue (SRR578947, SRR578948, SRR578941, and SRR578942) [2], *S. lycopersicum* leaf tissue (ERS2066781, ERS2067976, ERS2067977, ERS2066782, ERS2067978, ERS2067979) [16], *O. sativa* leaf tissue (SRR3503149, SRR3503151, SRR3503146, and SRR3503148) [21], and *P. patens* (GSM4218374- GSM4218393), were downloaded from NCBI SRA and processed as described above. To achieve comparable sequencing depths between

genotypes, we used seven million random reads from each of the public RNA-seq libraries. The genome assemblies used for *P. patens*, *A. thaliana*, *O. sativa* and *S. lycopersicum* were v3.3, TAIR10, v7.0 and 3.0, respectively.

## Annotations

*P.* patens gene and TE annotations (v3.3) were downloaded from CoGe https://genomevol-ution.org/coge/GenomeView.pl?gid=33928. *A. thaliana* TE annotation (TAIR10) was down-loaded from https://www.arabidopsis.org. *S. lycopersicum* TE annotation (Slycopersicu-m_514_ITAG3.2) was downloaded from Phytozome. *O. sativa* TEs were annotated de-novo using REPET software. TEs shorter than 100 bp were excluded for expression analysis. Intact LTR retrotransposons annotation (used in Fig 3B and 3C) was kindly provided by Prof. Stefan A. Rensing (University of Marburg, Germany).

## Supporting information

**S1 Fig. TE expression in Atddm1, Slddm1, Osddm1, Ppddm1 and Ppdnmt mutants.** A. Number of downregulated TEs in O. sativa, S. lycopersicum, A. thaliana and P. patens ddm1 mutants, as well as in P. patens dnmt mutants. B. Total number of RNA-seq reads mapped to indicated TE family types. Reads per million (RPM) values represent the average of replicates per genotype, error bars represent standard error.
(DOCX)

**S1 Table. Gene ID and accessions numbers.** Gene ID and accessions numbers for sequences used in evolutionary tree (Fig 1).
(DOCX)

**S2 Table. Summary of BS-seq data generated in this study.**
(DOCX)

**S3 Table. Summary of RNA-seq data generated in this study.**
(DOCX)

**S1 File. Up and down regulated TEs and genes in *Ppddm1*.**
(XLSX)

## Acknowledgments

We thank lab members that helped and supported the study.

## Author Contributions

**Conceptualization:** Nir Ohad, Assaf Zemach.

**Data curation:** Keith D. Harris, Assaf Zemach.

**Formal analysis:** Ofir Griess, Katherine Domb, Keith D. Harris, Assaf Zemach.

**Funding acquisition:** Nir Ohad, Assaf Zemach.

**Investigation:** Ofir Griess, Katherine Domb, Aviva Katz, Karina G. Heskiau, Nir Ohad, Assaf Zemach.

**Methodology:** Ofir Griess, Katherine Domb, Aviva Katz, Nir Ohad, Assaf Zemach.

**Resources:** Nir Ohad, Assaf Zemach.

**Supervision:** Nir Ohad, Assaf Zemach.

**Validation:** Katherine Domb, Aviva Katz, Karina G. Heskiau, Assaf Zemach.

**Visualization:** Ofir Griess, Katherine Domb, Assaf Zemach.

**Writing – original draft:** Ofir Griess, Nir Ohad, Assaf Zemach.

**Writing – review & editing:** Nir Ohad, Assaf Zemach.

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
