## [Decision Letter · Decision Letter 0]

18 Aug 2022

PONE-D-22-20689Knockout of DDM1 in Physcomitrium patens disrupts DNA methylation with a minute effect on transposon regulation and developmentPLOS ONE

Dear Dr. Zemach,

Thank you for submitting your manuscript to PLOS ONE. Your study has now been evaluated by two reviewers. As you will see both reviewers find your manuscript of potential interest but also raise a number of points that would need to be addressed before publication can be considered. Therefore, we invite you to submit a revised version of the manuscript that addresses the points raised during the review process. A decision on publication will then be made on the reviewers assessment of your revised version.

We look forward to receiving your revised manuscript.

Kind regards,

Anton Wutz

Academic Editor

PLOS ONE

Journal Requirements:

"This work was supported by the Israeli Centers for Research Excellence Program of the Planning and Budgeting Committee, Israel Science Foundation (757/12), Israel Science Foundation (1636/15), and the European Research Council (ERC, 679551) to A.Z, and Israel Science Foundation (767/09) to N.O."

"This work was supported by the Israeli Centers for Research Excellence Program of the Planning and Budgeting Committee, Israel Science Foundation (757/12), Israel Science Foundation (1636/15), and the European Research Council (ERC, 679551) to A.Z, and Israel Science Foundation (767/09) to N.O."

"This work was supported by the Israeli Centers for Research Excellence Program of the Planning and Budgeting Committee, Israel Science Foundation (757/12), Israel Science Foundation (1636/15), and the European Research Council (ERC, 679551) to A.Z, and Israel Science Foundation (767/09) to N.O."

Please state what role the funders took in the study.  If the funders had no role, please state: ""The funders had no role in study design, data collection and analysis, decision to publish, or preparation of the manuscript."" If this statement is not correct you must amend it as needed. 

5. Please amend your list of authors on the manuscript to ensure that each author is linked to an affiliation. Authors’ affiliations should reflect the institution where the work was done (if authors moved subsequently, you can also list the new affiliation stating “current affiliation:….” as necessary).

Reviewers' comments:

Reviewer's Responses to Questions

**Comments to the Author**

1. Is the manuscript technically sound, and do the data support the conclusions?

Reviewer #1: Yes

Reviewer #2: Partly

2. Has the statistical analysis been performed appropriately and rigorously? 

Reviewer #1: I Don't Know

Reviewer #2: No

3. Have the authors made all data underlying the findings in their manuscript fully available?

Reviewer #1: No

Reviewer #2: Yes

4. Is the manuscript presented in an intelligible fashion and written in standard English?

Reviewer #1: Yes

Reviewer #2: Yes

5. Review Comments to the Author

Reviewer #1: The manuscript by Griess et al., entitled “Knockout of DDM1 in Physcomitrium patens disrupts DNA methylation with a minute effect on transposon regulation and development”, describes the consequences of knocking out a gene in the moss ortholog to DDM1, well characterized as SNF2 chromatin remodelling protein in some flowering plants and there with a strong role in DNA methylation, transposon silencing, and plant development. The authors show convincingly that P.p. has only one DDM1 gene copy, which they successfully removed by gene replacement after homologous recombination. The resulting Ppddm1 mutant was analyzed for phenotypic changes, DNA methylation in all sequence context and transcriptome. The loss of PpDDM1 causes no obvious alteration of development, substantial and equal loss of mCG and mCHG, less loss of mCHH, but only a few upregulated TEs. Therefore, the consequences of a DDM1 loss-of-function are different from those in flowering plants, and these findings are interesting. The data are solid and well presented (some minor suggestions below).

Here are some points that could be considered in a revised version.

Figure 2 and 3: the representation of methylation differences between wt and ddm1 is counterintuitive. As the effects of the mutants are described, and ddm1 loses mC, positive values in panel 2B and 3B are irritating. Inverting the relation would help. Similarly, the mCHH hypermethylation at the small TE in panel 2G would also be easier to follow by subtracting wt from ddm1, not the other way. Panel Fig.2 I is difficult to understand, and the text does not make it clear what the conclusion should be.

In Fig. 2E and F, it would be good to indicate the position of the centromeres.

Figure 3 panel E: please include the data for the Ppmet and PpDNMT1 mutant.

A control for the completeness of the bisulfite conversion is missing. Also, it would be helpful to provide more information in the Supplemental Tables about the total read numbers, normalized read counts for the transcriptome data, and the alignment rates for all sequencing data.

What about the transcriptome analysis for the protein-coding genes? This was not discussed.

While the data are clear, their interpretation could have been extended and the Discussion could be more interesting. Could the authors speculate why the connection between DDM1, MET1, and CMT seems so different from that in Arabidopsis? And there is much more information from Arabidopsis about the connection between DDM1 and histone variants, which is ignored here but worth to discuss for P.p..

It would have made the manuscript even more interesting if the authors would have added data for a Ppddm1 complemented with a functional DDM1, either that from P.p. or from Arabidopsis, asking how much and which type of DNA methylation would be regained. But these experiments might be in progress.

Minor comments

Italicize gene and mutant names consistently.

Use Physcomitrium rather than Physcomitrella as genus name consistently (Methods part).

L. 59: what do mean by “nucleosomal DNA”? Nuclear? DNA in the context of chromatin?

L. 79: split the references: 2 and 3 refer to Arabidopsis, 25 to Physcomitrium.

L. 79: invert order: … difference between many flowering plants and P. patens is the robust CHG methylation in the latter, which …

L. 88: delete “with”.

L. 128: delete comma.

L. 142: replace “on” with “for”.

L. 197-198: make this sentence clearer

L. 216-217: sentence is incomplete

L. 337-339: make this sentence clearer

Figure 1 D-H: scale sizes are given in the legend, but bars are missing in all but one panel.

Figure 2 E and F: correct x-axis label to Chromosome 1.

Reviewer #2: Summary:

Greiss et al examined DNA methylation and mRNA expression in the moss Physcomitrium patens. They compared wild-type and a mutant of a ddm1 homolog in protonemata, which are gametophytic structures. In spite of a strong loss of methylation, they report that ddm1 has a minor effect on TE repression in protonemata, and no detectable morphological phenotype in protonemata nor in several other tissues. TEs that are upregulated in the ddm1 mutant are ones that were already expressed in wild-type, though at a lower level than in mutant. This manuscript has potential to make a valuable contribution toward understanding DNA methylation in a broader context beyond angiosperms. Of particular interest is the relationship between DDM1 and DNMT3 methyltransferase, a methyltransferase that does not exist in angiosperms.

Major concerns:

A central conclusion of the manuscriopt is the ddm1 has a weak effect on TEs in Physco relative to Angiosperms. However, it is not clear whether differences between the results here and prior results with angiosperms reflect differences in moss vs. angiosperms, or just between different tissue types. Could comparison of ddm1 mutant gametophyte vs sporophyte tissue in a single species also have a difference in the extent of TE activation? How about between two sporophytic tissues, for example ddm1 leaf vs floral bud in Arabidopsis? As the manuscript is currently written, it is hard to evaluate the significance of comparing haploid gametophyte moss tissue with diploid sporophyte angiosperm tissue.

An alternative explanation for the lack of detection of differentially expressed TEs in these experiments is poor quality sequencing libraries or insufficient coverage to detect poorly expressed TEs (even upon upregulation). These concerns could be addressed by demonstrating the quality of the sequencing libraries and by showing that the coverage was comparable to prior studies of other Physco mutants and of ddm1 mutants in other plants. Another possibility is that DDM1 effects are mainly limited to recently duplicated TEs. Such TEs are difficult or impossible to map using uniquely mapping RNA-seq reads to due to lack of sequence polymorphism. To address this, multi-mapping reads that allow comparisons to be made on a TE family basis rather than individual TE copies should be used. In this way, upregulated TE families can be identified even though you cannot say which individual copies are upregulated.

To be able to confidently interpret Figure 3A, the corresponding number of downregulated TEs and non-differentially expressed TEs should also be shown. Also need to indicate tissue source for each experiment, not just the species and mutant.

Moderate concern:

DDM1 appears to have opposite effects on CHH methylation at different loci. Since CHH methylation can be produced by both DNMT3 and by RdDM, and the relationship between DNMT3 and DDM1 is one of the more interesting aspects of this work, it could greatly strengthen the manuscript to define CHH loci as DNMT-dependent or RdDM-dependent. WT siRNA data (defines RdDM loci) and dnmt3 mutant methylation data (defines DNMT3 loci and by exclusion RdDM loci) are already available to do this.

Minor questions and concerns:

Regarding use of the word methylase: wouldn’t a methylase be an enzyme that cleaves methyl groups? For example, DNases cleave DNA, glycosylases cleave glycosyl groups. The correct word, I think, is methyltransferase, which is a methyl-adding enzymes.

What is the source of the H3 and H2K9me2 data in Figure 2J? Also, there are five violins per plot, but the legend indicates each violin represents a quartile. Is this a mistake?

Supplemental Table 1: Including transcript ID (isoform ID), not just gene ID would allow others to reproduce these trees.

How were TEs and gene annotations quality filtered for inclusion in BS-seq analysis?

Were protein trees built using global (end-to-end) or local alignment?

6. PLOS authors have the option to publish the peer review history of their article (what does this mean?). If published, this will include your full peer review and any attached files.

Reviewer #1: No

Reviewer #2: No

---

## [Author Response · Author response to Decision Letter 0]

7 Nov 2022

Comments from reviewer 1:

Comment 1: Figure 2 and 3: the representation of methylation differences between wt and ddm1 is counterintuitive. As the effects of the mutants are described, and ddm1 loses mC, positive values in panel 2B and 3B are irritating. Inverting the relation would help. Similarly, the mCHH hypermethylation at the small TE in panel 2G would also be easier to follow by subtracting wt from ddm1, not the other way. Panel Fig.2 I is difficult to understand, and the text does not make it clear what the conclusion should be. In Fig. 2E and F, it would be good to indicate the position of the centromeres. Figure 3 panel E: please include the data for the Ppmet and PpDNMT1 mutant.

As suggested we flipped the scales for figures 2B, 2G, 2J, 3B, and 3D. We edited Fig. 2I and edited the text to make it clearer. The location of centromeres in P. patens chromosomes has not been fully identified, yet, and is not so relevant to this study. The total hypomethylation in Ppmet is even lower than Ppcmt or Ppddm1. We focused mainly on Ppcmt because its hypomethylation effect is similar to that of ddm1 (including its CHH hypo/hyper methylation effect). 

Comment 2: A control for the completeness of the bisulfite conversion is missing. Also, it would be helpful to provide more information in the Supplemental Tables about the total read numbers, normalized read counts for the transcriptome data, and the alignment rates for all sequencing data.

We added sequencing stats to the Supplement (S2 and S3 Tables). 

Comment 3: What about the transcriptome analysis for the protein-coding genes? This was not discussed.

In comparison to TEs, the methylation effect on gene regulation is not necessarily direct. Because we could not find any obvious effect on gene regulation, we decided to leave it to future studies.

Comment 4: While the data are clear, their interpretation could have been extended and the Discussion could be more interesting. Could the authors speculate why the connection between DDM1, MET1, and CMT seems so different from that in Arabidopsis? And there is much more information from Arabidopsis about the connection between DDM1 and histone variants, which is ignored here but worth to discuss for P.p..

Thank you for your comment. The influence of PpDDM1 on CG and CHG methylation is not that different than that of AtDDM1. In both species, DDM1 is a strong regulator of CG and CHG methylation in heterochromatic TEs. The uniqueness of our finding is by showing that DDM1 can equally regulate CG and CHG methylation despite being targeted independently in P. patens. These findings are mentioned in the result and discussion sections. Since we did not profile histone modifications in Ppddm1, we decided not to draw any conclusions related to histone modifications.

Comment 5: It would have made the manuscript even more interesting if the authors would have added data for a Ppddm1 complemented with a functional DDM1, either that from P.p. or from Arabidopsis, asking how much and which type of DNA methylation would be regained. But these experiments might be in progress.

We agree that these experiments can be done in future studies.

Minor comments

Italicize gene and mutant names consistently. Done.

Use Physcomitrium rather than Physcomitrella as genus name consistently (Methods part). Done

L. 59: what do mean by “nucleosomal DNA”? Nuclear? DNA in the context of chromatin? A DNA that is wrapped around a histone octamer, i.e., a nucleosomal DNA (ref 14 that is cited there focuses on this aspect of DDM1).

L. 79: split the references: 2 and 3 refer to Arabidopsis, 25 to Physcomitrium. Done.

L. 79: invert order: … difference between many flowering plants and P. patens is the robust CHG methylation in the latter, which … Fixed.

L. 88: delete “with”. Deleted.

L. 128: delete comma. Deleted.

L. 142: replace “on” with “for”. We think that “on” is correct. 

L. 197-198: make this sentence clearer. Done.

L. 216-217: sentence is incomplete. Fixed.

L. 337-339: make this sentence clearer. Done.

Figure 1 D-H: scale sizes are given in the legend, but bars are missing in all but one panel. Added.

Figure 2 E and F: correct x-axis label to Chromosome 1. Fixed.

Comments from reviewer 2:

Comment 1: A central conclusion of the manuscriopt is the ddm1 has a weak effect on TEs in Physco relative to Angiosperms. However, it is not clear whether differences between the results here and prior results with angiosperms reflect differences in moss vs. angiosperms, or just between different tissue types. Could comparison of ddm1 mutant gametophyte vs sporophyte tissue in a single species also have a difference in the extent of TE activation? How about between two sporophytic tissues, for example ddm1 leaf vs floral bud in Arabidopsis? As the manuscript is currently written, it is hard to evaluate the significance of comparing haploid gametophyte moss tissue with diploid sporophyte angiosperm tissue.

We think that it is right to suggest that PpDDM1 has a different biological effect than that of published data in flowering plants for the following reasons. 

First, current data suggest that DNA methylation in flowering plants is necessary for silencing TEs in both sporophytic and gametophytic tissues. In fact, flowering plants have an active mechanism for protecting gametes from losing methylation and subsequent TE activation. While there are no expression profiles of ddm1 mutant in gametes of flowering plants, other mutants that cause to a slight hypomethylation in gametes were associated with TE activation (Borg et al., Elife. 2021, Long et al., Science 2021, He et al., Elife. 2019). Therefore, it is unreasonable to assume that ploidy is the reason for the trivial TE effect in Ppddm1. 

Second, P. patens is different from flowering plants in many ways, ploidy cycle is only one aspect. For example, distribution of genes and TEs along chromosomes is very different between P. patens and flowering plants, the missing of gene body methylation is another unique feature of bryophytes and P. patens, and of course the types of DNA methylases in P. patens and flowering plants are different. While we cannot control all different elements between P. patens and flowering plants, we can compare the TE expression effect in ddm1 mutants to the potential of TE expression derived from DNA methylation mutants. Accordingly, by showing the trivial TE activation effect of Ppddm1 versus the robust activation in Ppdnmt mutants (derived from the same tissue), we think that it is right to suggest for a weak effect of PpDDM1 mutation on TE expression, which is different than published data on ddm1 in flowering plants.

Saying that, to be extra careful with our claims and to accommodate the reviewer’s comment, we made the following edits: 1) we have replaced “show” by “suggest” in the last sentence of the abstract. 2) In the discussion we added the underlined text in the following sentence “In comparison to published data in flowering plants, our data show that mutation of ddm1 in moss, does not cause a substantial activation of TEs (at least not in the protonema tissue) nor an obvious developmental phenotype.”

Finally, we would like to note that our main findings on the effect of PpDDM1 on DNA methylation, TE expression, and development, are novel and unique by themselves and therefore warrant publication. The title and most of the manuscript represent these main findings. The comparison to flowering plants is only secondary to our main findings and as explained above, is based on reasonable assumptions, which we therefore think that it can be suggested in the manuscript.

Comment 2: An alternative explanation for the lack of detection of differentially expressed TEs in these experiments is poor quality sequencing libraries or insufficient coverage to detect poorly expressed TEs (even upon upregulation). These concerns could be addressed by demonstrating the quality of the sequencing libraries and by showing that the coverage was comparable to prior studies of other Physco mutants and of ddm1 mutants in other plants. Another possibility is that DDM1 effects are mainly limited to recently duplicated TEs. Such TEs are difficult or impossible to map using uniquely mapping RNA-seq reads to due to lack of sequence polymorphism. To address this, multi-mapping reads that allow comparisons to be made on a TE family basis rather than individual TE copies should be used. In this way, upregulated TE families can be identified even though you cannot say which individual copies are upregulated.

We agree with the reviewer that coverage is a valid argument. We added RNA-seq stats in S3 Table. Mapping rate of the RNA was 86% for all samples, indicating its high quality. The coverage of our RNA-seq libraries were about 7M reads per replicate, which is generally okay but lower than most analyzed samples. Accordingly, to normalize for library size we reanalyzed all samples by using only 7M reads per replicate. While these analyses reduced the number of upregulated TEs in the different samples (shown in the updated Figure 3A), the difference to Ppddm1 was still substantial and therefore did change our original conclusions.

Meta-expression analysis per TE family type is another great suggestion raised by Reviewer 1. We did this analysis (S1B Fig), however, did not discover any substantial TE expression in Ppddm1 there either.

We thank the reviewer for raising these excellent points that helped to increase the confidence of our results.

Comment 3: To be able to confidently interpret Figure 3A, the corresponding number of downregulated TEs and non-differentially expressed TEs should also be shown. Also need to indicate tissue source for each experiment, not just the species and mutant.

In the field of DNA methylation, scientists usually ignore the negligence number of downregulated TEs. However, we accept the reviewer request and included those numbers in S1A Fig and mentioned it in the relevant results section. 

Comment 4: DDM1 appears to have opposite effects on CHH methylation at different loci. Since CHH methylation can be produced by both DNMT3 and by RdDM, and the relationship between DNMT3 and DDM1 is one of the more interesting aspects of this work, it could greatly strengthen the manuscript to define CHH loci as DNMT-dependent or RdDM-dependent. WT siRNA data (defines RdDM loci) and dnmt3 mutant methylation data (defines DNMT3 loci and by exclusion RdDM loci) are already available to do this.

DNMT3 is responsible for almost the entire CHH methylation in P. patens. In the manuscript, we show the strongest connections we found between CHH hypermethylation and chromatin features. 

Comment 5: Regarding use of the word methylase: wouldn’t a methylase be an enzyme that cleaves methyl groups? For example, DNases cleave DNA, glycosylases cleave glycosyl groups. The correct word, I think, is methyltransferase, which is a methyl-adding enzymes.

‘Ase’ is a suffix for enzymes in general. In the Methylation field, we use ‘methylase’ and ‘demethylase’ for enzymes that methylate and demethylate DNA or other substrates (e.g., histones), respectively.

Comment 6: What is the source of the H3 and H2K9me2 data in Figure 2J? Also, there are five violins per plot, but the legend indicates each violin represents a quartile. Is this a mistake?

Legend indicates quantile. 

Comment 7: Supplemental Table 1: Including transcript ID (isoform ID), not just gene ID would allow others to reproduce these trees.

Gene models were added to the S1 Table.

Comment 8: How were TEs and gene annotations quality filtered for inclusion in BS-seq analysis?

Explanation about gene and TE annotations is provided in the Methods section.

Comment 9: Were protein trees built using global (end-to-end) or local alignment?

Global, as conserved regions are found throughout the protein.

---

## [Decision Letter · Decision Letter 1]

6 Dec 2022

PONE-D-22-20689R1Knockout of DDM1 in Physcomitrium patens disrupts DNA methylation with a minute effect on transposon regulation and developmentPLOS ONE

Dear Dr. Zemach,

Thank you for submitting the revised version of your manuscript to PLOS ONE. Your revision has now been evaluated by the two original reviewers. As you will see both reviewers find your study has been stengthened and all concerns have been addressed. I return your manuscript to you for a last round of revisions for you to consider the remaining points of the reviewers. If you have addressed the remaining points and send a point-to-point response along with your further revised manuscript, I would be in a strong position to make a decision on publication of your study.

We look forward to receiving your revised manuscript.

Kind regards,

Anton Wutz

Academic Editor

PLOS ONE

Journal Requirements:

Reviewers' comments:

Reviewer's Responses to Questions

**Comments to the Author**

1. If the authors have adequately addressed your comments raised in a previous round of review and you feel that this manuscript is now acceptable for publication, you may indicate that here to bypass the “Comments to the Author” section, enter your conflict of interest statement in the “Confidential to Editor” section, and submit your "Accept" recommendation.

Reviewer #1: (No Response)

Reviewer #2: (No Response)

2. Is the manuscript technically sound, and do the data support the conclusions?

Reviewer #1: Yes

Reviewer #2: Yes

3. Has the statistical analysis been performed appropriately and rigorously? 

Reviewer #1: Yes

Reviewer #2: Yes

4. Have the authors made all data underlying the findings in their manuscript fully available?

Reviewer #1: Yes

Reviewer #2: Yes

5. Is the manuscript presented in an intelligible fashion and written in standard English?

Reviewer #1: Yes

Reviewer #2: Yes

6. Review Comments to the Author

Reviewer #1: In the revised version of the manuscript by Griess et al., entitled “Knockout of DDM1 in Physcomitrium patens disrupts DNA methylation with a minute effect on transposon regulation and development”, the authors have addressed most of the points raised during the initial review. The different representation of the methylation changes makes it now more logic. Several other points were answered in the response of the authors. My suggestion to make the Discussion more interesting was not followed but that is a matter of taste. I just do not understand why the data for the other Pp mutants were not added to Fig. 3E. The authors write that the total hypomethylation in Ppmet is even lower than Ppcmt or Ppddm1, but this is no argument not to include them. It would help to gain a more complete picture of the effect of all mutants.

There are a few minor corrections necessary:

Line 211: replace “imply for” with “implies”

Figure 2 E and F: replace 2x “chromsome” with “chromosome”

Figure 2 F: change ddm1 to italic

Reviewer #2: All my concerns have been adequately addressed except that I don't see the source or analysis methods for the H3 and H2K9me2 data in Figure 2J.

7. PLOS authors have the option to publish the peer review history of their article (what does this mean?). If published, this will include your full peer review and any attached files.

Reviewer #1: No

Reviewer #2: **Yes: **Jonathan Gent

---

## [Author Response · Author response to Decision Letter 1]

9 Dec 2022

Response to Reviewers

Reviewer #1: In the revised version of the manuscript by Griess et al., entitled “Knockout of DDM1 in Physcomitrium patens disrupts DNA methylation with a minute effect on transposon regulation and development”, the authors have addressed most of the points raised during the initial review. The different representation of the methylation changes makes it now more logic. Several other points were answered in the response of the authors. My suggestion to make the Discussion more interesting was not followed but that is a matter of taste. I just do not understand why the data for the other Pp mutants were not added to Fig. 3E. The authors write that the total hypomethylation in Ppmet is even lower than Ppcmt or Ppddm1, but this is no argument not to include them. It would help to gain a more complete picture of the effect of all mutants.

We added methylation mutants (including Ppmet) to Figure 3E.

There are a few minor corrections necessary:

Line 211: replace “imply for” with “implies”

Replaced.

Figure 2 E and F: replace 2x “chromsome” with “chromosome”

Replaced.

Figure 2 F: change ddm1 to italic

Changed.

Reviewer #2: All my concerns have been adequately addressed except that I don't see the source or analysis methods for the H3 and H2K9me2 data in Figure 2J.

The reference for Histone H3 and H3K9me2 was added to figure legends.

---

## [Editor Report · Decision Letter 2]

13 Dec 2022

Knockout of DDM1 in Physcomitrium patens disrupts DNA methylation with a minute effect on transposon regulation and development

PONE-D-22-20689R2

Dear Dr. Zemach,

thank you for sending the further revised version of your study, which addresses all remaining points of the reviewers in a satisfactory manner. Therefore, I find your study is now suitable for publication.

We’re pleased to inform you that your manuscript has been judged scientifically suitable for publication and will be formally accepted for publication once it meets all outstanding technical requirements.

Kind regards,

Anton Wutz

Academic Editor

PLOS ONE
---

## [Editor Report · Acceptance letter]

10 Feb 2023

PONE-D-22-20689R2 

Knockout of DDM1 in *Physcomitrium patens* disrupts DNA methylation with a minute effect on transposon regulation and development 

Dear Dr. Zemach:

I'm pleased to inform you that your manuscript has been deemed suitable for publication in PLOS ONE. Congratulations! Your manuscript is now with our production department. 

Kind regards, 

on behalf of

Dr. Anton Wutz 

Academic Editor

PLOS ONE